# Are You My Host? An Overview of Methods Used to Link Bacteriophages with Hosts

**DOI:** 10.3390/v17010065

**Published:** 2025-01-05

**Authors:** Paul Hyman

**Affiliations:** Department of Biology and Toxicology, Ashland University, Ashland, OH 44805, USA; phyman@ashland.edu

**Keywords:** bacteriophage, host range, host prediction, metagenomics, viromics, phage therapy

## Abstract

Until recently, the only methods for finding out if a particular strain or species of bacteria could be a host for a particular bacteriophage was to see if the bacteriophage could infect that bacterium and kill it, releasing progeny phages. Establishing the host range of a bacteriophage thus meant infecting many different bacteria and seeing if the phage could kill each one. Detection of bacterial killing can be achieved on solid media (plaques, spots) or broth (culture clearing). More recently, additional methods to link phages and hosts have been developed. These include methods to show phage genome entry into host cells (e.g., PhageFISH); proximity of phage and host genomes (e.g., proximity ligation, polonies, viral tagging); and analysis of genomes and metagenomes (e.g., CRISPR spacer analysis, metagenomic co-occurrence). These methods have advantages and disadvantages. They also are not measuring the same interactions. Host range can be divided into multiple host ranges, each defined by how far the phage can progress in the infection cycle. For example, the ability to effect genome entry (penetrative host range) is different than the ability to produce progeny (productive host range). These different host ranges reflect bacterial defense mechanisms that block phage growth and development at various stages in the infection cycle. Here, I present a comparison of the various methods used to identify bacteriophage-host relationships with a focus on what type of host range is being measured or predicted.

## 1. Introduction: What Is Host Range and Why Would Anyone Care About It?

Host range can be described very simply as the types or range of host bacteria that a particular bacteriophage can infect. For some phages, this reflects the number of strains of a single species of bacteria that can be infected, but some phages can infect multiple species and even species in several genera, families, etc. Historically, host range was considered to be principally due to receptor–receptor binding protein interactions [1]. But as knowledge of bacteria–phage interactions, especially bacterial anti-phage (defense) mechanisms, has grown, it is clear that host range can be affected at many points during the infection process [2,3,4,5,6]. In 2010, Steve Abedon and I identified multiple types of host range depending on the effect on the bacterial host and the stage of infection where a resistance mechanism acts [2]. Among others, these include the following:
Adsorptive host range: the phage can reversibly and irreversibly bind to the bacterial host.Penetrative host range: the phage is able to insert its genome into the host cell.Bactericidal host range: the phage can kill the bacterial host cell without regard to phage progeny production.Productive host range: progeny phages are produced, usually, but not necessarily, killing the host.Plaquing host range: the progeny phages are able to form plaques on lawns of host bacteria on solid media.

A key differentiator among these host ranges is the understanding that phage binding to a cell does not guarantee phage genome entry into a cell and that the latter does not guarantee production of progeny phage or bacterial host death. Various types of bacteria have evolved anti-phage mechanisms that can interrupt the phage life cycle at different stages of the cycle. Archaea also contain many of these mechanisms as protection against infection by archaeal viruses. Overall, these defenses are being likened to eukaryotic defense systems, including the animal immune system as an antiviral mechanism. Some of these bacterial systems, like restriction enzymes and CRISPR-Cas systems, are widely distributed among bacteria while others, such as specific toxin-antitoxin systems, are more limited in distribution [7].

The host range of a bacteriophage has been described as narrow or broad although these terms are problematic, as there are no standards that define them [8,9]. Instead, these terms are best used as comparative adjectives between phages that have been tested against the same bank of hosts. Broadly speaking, however, it is clear that some phages seem to infect many host strains while some only infect a few [10,11]. This has implications both for phage biology and for applications of phages. In models of microbial ecology, for example, phages that infect multiple species can be considered to be generalist feeders, even if the infection efficiency is not the same for all hosts [6,12]. Phages that infect just a few types of bacteria would be described as specialist feeders. Both may be found in the same environments acting as bacterial predators, but the generalists would likely play a greater role in, for instance, pathways of horizontal gene transfer between species.

The various host ranges listed above are applicable to different areas of phage research. For those interested in phage therapy, the use of bacteriophages as biocontrol agents of typically pathogenic bacteria, bactericidal, productive, or plaquing host ranges are the most relevant as they all demonstrate killing of bacteria [13,14]. The goal of testing is to show that a particular phage (or mixture of phages) can kill the pathogenic bacteria infecting the patient(s). In contrast, researchers studying population-level phage biology, such as the movement of mobile genetic elements through populations, will be more interested in the penetrative host range of bacteriophages that are capable of facilitating transduction [2,15]. The penetrative host range is broader than the productive host range because, while all phages that kill a bacterium must insert their genome into the host cell, not all phages that have a phage genome enter the host cell will kill the host cell. These nonproductive infections may be due to protection by antiphage mechanisms such as restriction enzymes, CRISPR systems, and toxin-antitoxin proteins.

Another consideration when identifying bacteriophage hosts is the ultimate goal of the identification. As described below, there are both culture-based and non-culture-based host identification systems. These differ in their utility for follow-up studies. If there is a desire for further study of additional phage–host interactions beyond just identification, both the bacteriophage and, especially, its host bacteria, must be isolated and able to be grown in the laboratory.

Until recently, host range testing has been accomplished using culture-based systems, but metagenomic methods have revealed the diversity of the unculturable microbes in aquatic [16,17] and terrestrial [18,19] environments. This has created the need to match bacteriophages from the virome portion of a metagenomic data set to the bacteria they might infect, and has led to the development of in silico methods of predicting matches of phages and their hosts. Host identification in these data sets is not necessarily at the species or strain level, but rather at higher taxonomic levels [20]. These methods also differ from culture-based methods in that they do not usually predict a productive host range, but rather identify a particular host genus or species that matches a specific phage genome and is likely to have some strains that the phage can infect [21,22].

At this time, a variety of methods have been proposed for identifying and predicting host range and matching individual phages to hosts. These range from simple spot testing to metagenomic methods that link phages and hosts. Most recently, methods that rely on proximity (spatial co-localization) of phages and host genomes have been developed. In this review, I provide an overview of these types of methods, along with some detailed descriptions of each method.

## 2. Overview of Methods

Methods for determining or predicting which hosts a phage can infect can be divided into three major groups: growth-based methods, genome sequence-based methods, and proximity-based methods (Table 1). The first group uses cultured phage and hosts, while the latter two are based on the sequencing of individual phages and hosts, either as environmental sample metagenomes and/or metatranscriptomes or as isolated (but not cultured) phage–host pairs. Growth-based methods are used after phage isolation, while the other methods do not require phage isolation and are, in fact, often incompatible with the isolation of phages or hosts. Some of the non-growth-based methods are also predictive of possible hosts, rather than actually demonstrating that a phage can infect a particular host. Non-growth-based methods have applications in understanding community level interactions of phages and hosts, as well as answering other ecological questions. For example, metagenomic methods were developed to study unculturable microbes in bulk, not as individual strains. Likewise, matching phages and hosts in an environmental sample using metagenomic data may also identify broader categories of hosts, rather than individual strains. Proximity-based methods, however, can include sequencing of single cells after separating phage–host pairs into single units to analyze, but can, at best, only suggest production of progeny phages from these hosts. These methods can answer questions that cannot be studied with cultured phages and hosts, but it is important to recognize that each method addresses different aspects of phage–host biology. For other reviews on methods and the questions they can answer, see [23,24,25].

### 2.1. Who Can You Kill? Growth-Based Methods

Spot testing, plaque testing, culture clearing—these are the “classical” methods of showing phage killing of particular hosts used in individual phage characterization as well as selecting phages for phage therapy. They all rely on bacteria being killed by the phages added to the bacteria on solid or liquid media. These methods are also used when the phages are isolated from environmental samples on specific hosts. Subsequently, the newly isolated bacteriophages can be tested for killing other potential hosts. Groups of new isolated phages typically have a mix of narrow and broader host ranges [9,10], and host range testing is needed to determine which phages have which host range.

These methods involve testing isolated phages against a series of culturable hosts, essentially one at a time, although some high-throughput methods have been developed [26,27]. This means that both phages and hosts must be culturable in the lab. While spot testing and culture clearing are prone to false positives from other killing agents in the phage stock, such as lysin proteins from the infections that produced the stock (especially in unpurified phage stocks), their simplicity keeps them in widespread use. An additional benefit of culture clearing methods is that they can be accomplished in multi-well plates monitored using plate readers allowing for high-throughput testing. A loss of turbidity in the culture is interpreted as killing of the hosts [28]. One limitation to culture testing is the need to frequently or continuously monitor the culture turbidity because bacteriophage-resistant bacteria will often appear in the culture. A decrease in turbidity, often transitory, is interpreted as a positive result, rather than a culture that remains cleared. For many bacteria, this may require situating a plate reader in an incubator or using a plate reader that can also maintain an elevated temperature.

Depending on the specific method used, growth-based methods may be determining bactericidal, productive, or plaquing host range. Bactericidal host range can be assessed by spot testing or culture clearing. For both of these, relatively large numbers (10^3^ or more) of phages are commonly applied. With dilution of phage stocks, the production of plaques on solid media or limiting dilution to a single phage for broth clearing can be assessed, which will reflect a phage’s plaquing or productive host range, respectively.

PhageFISH (FISH—fluorescence in situ hybridization) is a newer, more technically challenging, method that also relies on phage growth in host cells [29]. This method is based on the visualization of phage genomes and host cells with fluorescent labels. These labels are commonly labeled nucleic acid probes, so at least part of the phage genome sequence must be known. Co-localization of the phage genome signal and bacterial cell signal is generally interpreted as indicating infection of the cell. If many copies of the phage genome are detected in a single cell, this is interpreted as showing phage replication. Strictly speaking, this is not completely equivalent to productive host range, as abortive infection mechanisms may block progeny production by destroying the cell before the phage infection cycle is complete. In addition, this method is laborious and better suited for studying the phage infection process than for determining host range.

### 2.2. Who Do You Look Like? Genome Sequence-Based Methods

These methods are often applied to metagenomic data that sample entire communities, most of which can consist of bacteria and phages that cannot be grown in culture. There is a wide variety of analyses used, but all rely on comparisons of particular features, such as sequence similarities in phage genomes to CRISPR spacer sequences; tRNA genes in phage genomes; sequence motifs, such as tetranucleotide frequencies; and marker proteins, such as RNA polymerases or structural proteins [20,30,31]. These methods are sometimes predictive of possible hosts, rather than demonstrating the ability of a phage to infect a particular strain of a particular host species as growth-based methods do.

Phage sequences and other genomic features may also be compared to those of phages with a known host range and similar features [32,33]. For phages, multiple elements are used because there is not a universal gene or sequence found in all phages equivalent to the ribosomal RNA genes of cellular organisms. Broadly, the various sequence elements are compared to databases of other sequences. This is both the strength and limitation of these methods. While the databases contain millions of sequences, they are not all equivalent, and the same analysis using different reference databases can yield varying results [20]. These methods can also be applied to viruses of eukaryotic microbes [34]. One challenge to these methods is the mosaic nature of phage evolution. Two phage species may be a mix of identical and disparate regions, making relationships more difficulty to determine.

A dizzying number of software packages and analytical pipelines have been developed for analyzing metagenomic data, extracting data relevant to matching phages and bacterial sequences, and using databases to predict hosts from phage genome sequences. Overviews of these can be found in [20,35,36,37,38].

While most software directly analyzes sequences, some researchers have used either sequence databases or features analysis to create learning databases for machine learning programs. Machine learning is a specialized version of artificial intelligence, in which software uses datasets and statistical analysis to make predictions (https://www.ibm.com/topics/machine-learning, accessed on 30 December 2024). Machine learning tools have been used to predict a number of phage parameters, including lifestyle (lytic vs. temperate) and phage–host matching (see [36] for a review of these studies). For example, Wang and colleagues [39] developed a machine learning net that was trained using sequence and host data from over 800 phages listed in the NCBI REfSeq database. Once trained, they used the net software to predict the host for crAss001 phage and for phages in two metagenomic datasets. They found that their results were consistent with the predictions of more specialized software. A similar approach was used by Amgarten and colleagues [33], who used a database of phage genomes and annotated protein sequences to train a neural network-based analysis tool to predict hosts either at the genus or species level. Yang and colleagues [40] used a machine learning net-based tool they designated SpikeHunter to identify tailspike receptor binding proteins in a phage database. They then identified a set of tailspike proteins in prophages found in bacterial genomes. Because prophages must have infected the host cell they are in, the prophage tailspike sequences could be used to predict the serotype of bacteria the phages would infect.

A different computational method that may be able to identify phage–host relationships in the future are neural network-based protein structure prediction systems, such as AlphaFold [41,42]. A neural network is a specialized form of machine learning that is well suited for complex pattern recognition (https://www.ibm.com/think/topics/ai-vs-machine-learning-vs-deep-learning-vs-neural-networks, accessed on 30 December 2024), such as the recognition of protein structural motifs. While Alphafold is currently used to identify proteins from sequence information [43,44], improved prediction of the structure of the receptor binding domain of receptor-binding proteins may lead to predictions of the receptor and what potential host cells make that receptor on their surfaces. This will likely also require improved datasets of receptor-binding proteins matched to their corresponding receptors.

While most sequence-based methods cannot predict hosts below the species or genus level, Gencay and colleagues [45], using a collection of 41 newly isolated phages and 71 environmental strains of Salmonella, found that they could use statistical analysis to identify which phage characteristics were the most predictive of the host range among the strains of Salmonella. They used a combination of genomic and phenotypic characteristics in their analysis and found that the host range of the phages most strongly correlated with the genus of the phage and the molecule that acted on the bacterial surface as the receptor. Recently, Gaborieau and colleagues [46] analyzed the phage–host interactions of 403 *E. coli* strains and 96 phages. The predictability of whether a phage could infect a host based on phage adsorption to the host cell was 86%. They further used this dataset to train a machine learning algorithm that designed phage cocktails. In contrast to these two studies, Piel and colleagues [47] isolated 195 *Vibrio crassostreae* strains along with 243 vibriophages from an oyster farm. Their analysis of phage–host interactions found that host defense systems defined whether groups of hosts were susceptible to phages. Together, these studies demonstrate that statistical analysis of a broader combination of traits can be useful, but it is unclear how generalizable these results are. Each one focused on strains of a single species, so it is uncertain if these methods can be applied to less well-defined species of bacteria.

There are limitations to sequence-based methods. If the original sample is virion particles separated from cells, rather than total environmental DNA, any phage-infected cells or integrated prophages will be excluded, along with the other cells in the sample, biasing any results toward virion particles [48]. Methods that rely on comparing protospacer sequences in CRISPR-Cas arrays to phages will only be usable with hosts that encode CRISPR-Cas systems that have been estimated to be between 10% [49] and 40% [50] of bacteria. Finally, the various software tools are better at predicting hosts at the genus or high taxonomic levels (see Table 1 references).

### 2.3. Who Are You Near? Proximity-Based Methods

Proximity-based methods all attempt to capture phages that are attached to or already infecting a host cell. Methods such as viral tagging and AdsorpSeq sequence both the phage and bacterium using single-cell sequencing methods, while others such as microfluidic digital PCR and EpicPCR use PCR of marker genes to establish identity or relationships of bacterium and phage (see Table 1 and Figure 1 for details of each method). The capture methods vary, including FACS (fluorescence-activated cell sorting), which physically separates phage-bound bacteria from unbound bacteria; microdroplets that can contain a single phage–host pair; solid matrices for separating and containing phage–host pairs, and chemical cross-linking of phage and host DNA. In all cases, they take advantage of the phages’ natural affinity to bind to a bacterium that is likely within the phages’ host range and to inject the phage genome into the host.

Some of these methods are more effective when the phage or host bacteria can be cultured and then used as “bait” to attract their partner. Viral tagging, for example, was first used to quantify hosts of particular viruses in mixtures of bacteria [51]. In contrast, Unterer and colleagues [52] used FACS or microfluidic droplets to separate infected cells from uninfected and measure some phage growth dynamics. While they used known phages and hosts to demonstrate single infected cell isolation, this method could be used to capture uncultivated phages from an environmental sample. Other methods, such as microfluidic digital PCR or EpicPCR, can be used on uncultured hosts and phages. Both phage and bacteria need to be somewhat related to known species to create degenerate PCR primers that can be used to identify the host via 16S gene targeting and the phage via a phage marker gene.

More recently, the chromosome conformation capture method (Hi-C, also called proximity ligation or meta3C) has been developed to bypass the need for any knowledge of phage or bacterial sequences. In this method, DNA in a single cell is chemically cross-linked so that any parts of the DNA that are in close proximity may be cross-linked together. In a phage-infected cell, this can lead to cross-linking of phage and host genome segments. The DNA is then cut with a restriction enzyme and rejoined using DNA ligase. DNA molecules in close proximity, including host–phage hybrid molecules, are then sequenced and analyzed for identification [53,54]. Various software pipelines can be used to analyze the resulting DNA sequences. Du and colleagues [55] have developed software with improved selection of phage–host hybrid molecules.

Proximity-based methods have the advantage of bypassing a need to culture the phages or the hosts. However, these methods do not directly demonstrate progeny production. Instead, they are identifying hosts that are within the phages’ adsorptive or penetrative host range. Environments with high levels of phages may produce false positives, as phages may be captured that are near the host but not bound to it. Phage may also be detached from hosts if they are not irreversibly bound and fail to produce a positive signal associating the phage and the host. For methods that use a marker gene rather than whole genome sequencing (e.g., microfluidic digital PCR; iPolony; epicPCR), only phages with a marker gene recognized by the degenerate PCR primers will be detected. Unknown phages will be missed. A final difficulty of all these methods is that they are technically challenging and are dependent on the efficiency of multiple steps.

**Table 1 viruses-17-00065-t001:** Methods used to link bacteriophages with host bacteria.

	Description	Question Being Answered (Phage POV)	Challenges	Host Range Described	References
Culture-based methods (requires host, and usually phage, to be grown in the lab)					
Spot testing	Small droplets of phage stock, often up to 10^7^ pfu, or environmental culture are placed in a 1–10 µL drop onto a plate with a lawn of bacteria. After incubation, a zone of bacterial killing indicates presence of a phage or phages able to kill that host.	Who can I kill?	Prone to false positives from other bactericidal substances; difficult to scaleup.	Bactericidal.	[13,56,57,58,59]
Confluent lysis method (routine test dilution)	Serial dilutions of phage stocks are added to a lawn of bacteria. After incubation, dilution that nearly causes confluent lysis is scored.	Who can I kill?	Same as spot testing.	Bactericidal.	
Plaque testing	Serial dilutions of a phage stock or other sample are placed onto a lawn of bacteria in 1–10 µL drops. After incubation, plaques are observed at greater dilutions. Alternatively, dilutions of phage can be mixed with bacteria in soft agar and poured on individual plates.	Who can I produce progeny on?	Not all bacteriophages can form plaques under standard growth conditions; difficult to scaleup.	Plaquing.	
Broth clearing	Similar to plaque testing, except that aliquots of serial diluted phage are added to broth cultures of bacteria that cannot be plated. Culture clearing (prevention of turbidity development) indicate that phages are killing the host bacteria.	Who can I kill?	Prone to false positives from other bactericidal substances; timing dependent as resistant hosts will appear after initial clearing causing false negatives; scaleup is possible with incubator-plate readers but capacity is still limited.	Bactericidal.	
PhageFISH	This method uses a host-specific RNA probe and phage-specific probes to detect both genomes in situ. Microscopic observation of co-localization of phage genome and host genome indicates penetration. Increasing numbers of phage genomes inside of cell indicates progeny production.	Who can I infect? Who can I produce progeny on?	Technically difficult, requires at least partial genome sequence of phage, difficult to scaleup.	Penetrative and productive.	[29,60]
Non-culture-based methods (can be used on environmental samples but may be limited to phages or hosts that are closely enough related to a reference strain so that PCR primers or probes will work, or sequence comparisons can be made)					
Sequencing-based					
Genome sequencing methods—short read, long read and combined results	Metagenomes can be used to compare unknown phages to known ones by simple sequence matching such as using Blast or by more complex comparisons of key genes, protein families, or multiple other features.	Who do my gene sequences resemble?	All sequencing methods are sensitive to the database used for comparison; can confuse prophages and active infections; host prediction accuracy decreases at higher specificity (i.e., at genus or species level relative to order or phylum); when using marker genes, phage mosaicism may lead to false positives.	Productive (inferred for ancestors of identified phages).	[32,33,35]
	Combination of metagenomic and single cell sequencing data; single cell-virus co-occurrence interpreted as infection.	Who do my gene sequences resemble? Who am I found in?		Productive and penetrative.	[61]
Metagenomic co-occurrence network	Metagenomic assembly of multiple samples analyzed for co-occurrence of virus and host.	Who am I found with?	Matching at genus level only; accuracy varied from 57 to 87%, depending on stringency of scoring.	Unclear.	[62]
CRISPR spacer sequencing	Software identifies CRISPR array and compares CRISPR spacers to known phages in databases.	Who had ancestors that survived being infected by my ancestor?	Not all bacteria use a CRISPR system; spacer sequences are replaced after more recent phage encounters, only identifies phages (or near relatives) that have been sequenced for matching.	Penetrative.	[63,64]
	A combination of CRISPR spacer and tRNA analysis.	Who had ancestors that my ancestors sometimes, but not always, successfully infected.	It can identify a minority of phages to genus or species, but most are at higher taxonomic levels, and some are not matched; matching is database dependent.	Productive (inferred from persistence of phages over time).	[31]
Proximity-based methods					
Viral tagging	Viruses (either cultured or environmental) are labeled with a fluorescent stain and mixed with bacteria. Early versions of this technique only showed binding by phage via microscopic observation, but later versions added FACS sorting to isolate phage–host pairs that could be identified by sequencing.	Who can I bind to?	Specific virus–host identification requires FACS instrument, single-cell genomic sequencing.	Adsorptive.	[65,66]
AdsorpSeq	Similar to viral tagging, except that phage–host pairs are isolated by differential migration through an agarose gel.	Who can I bind to?	Used with a known host to collect phages from an environmental sample, so it cannot be used with non-culturable hosts.	Adsorptive.	[67]
Microfluidic digital PCR	Degenerate primers for phage marker gene and host 16S RNA genes amplified in single-cell drops.	Who am I found near or infecting?	False positives due to phage, host capture in droplet without infection; technically difficult.	Adsorptive or penetrative.	[68]
Polony (iPolony) method	The polony method is used for viral enumeration by fixing DNA at distinct points on solid matrix and performing local amplification for visualization. The infected cell polony (iPolony) method uses flow cytometry to count and separate cells by some groups’ features (e.g., containing chlorophyll). The cells are embedded in a matrix and the infected cells are identified by the polony method. Group-specific probes can differentiate groups of phages.	How many cells am I infecting?	Limited to closely related phages, so that degenerate primers will amplify marker gene; technically difficult.	Not useful for host matching or host range, but more focused on population level infections.	[69,70,71]
EpicPCR	The phage marker gene and host 16S gene amplified in droplets and then joined by fusion PCR. Sequencing is used to show linkage.	Who am I found near or infecting?	False positives due to phage, host capture without infection or DNA escape from one droplet to another; only identifies phage with target gene; technically difficult.	Adsorptive or penetrative.	[72,73,74]
Hi-C (chromosome conformation capture, also known as proximity ligation)	Genomic DNAs (host and phage) are cross-linked chemically, then digested with a restriction enzyme and ligated. DNAs in proximity will be linked together, including phage and host hybrid fragments. Analysis of short read sequencing of ligated DNA reveals hybrid fragments. Sometimes combined with long read sequencing.	Who can I infect?	False positives due to phage; host proximity without infection; can only detect viruses with some homology to viruses already sequenced (genomic or viromic); low sensitivity for low abundance phages; technically difficult.	Adsorptive or penetrative. (Productive has been inferred when phage genome is over-represented in sequences (i.e., there are many copies of the genome in a cell).	[53,54,74,75,76,77,78]

## 3. Conclusions

Host range, while relatively simple in concept, arises from multiple mechanisms and interactions between bacteriophages and bacteria. In designing studies of host range and host prediction, the technique should be appropriate to the problem under study, as different methods produce results describing different host ranges. Of the three broad groups of methods discussed here, the culture-based methods are most appropriate for host killing or progeny production testing, such as in phage therapy. Culture-based methods have not changed substantially since they were first developed decades ago and remain the gold standard for demonstrating bacterial killing. Sequence-based methods and proximity-based methods are newer and have a distinct advantage in being applicable to non-culturable species. They also have a rich potential for further development to become more accurate and detailed in their predictions. This is likely for several reasons. Both groups utilize sequence analysis to some extent, and sequencing technologies, especially single-cell sequencing, continue to improve. Both groups are also often dependent on databases of known phage–host interactions. These data should also continue to become more abundant as each new study adds to the databases. Finally, the application of computational methods, such as neural networks and artificial intelligence, is just beginning in host prediction. Together, these trends suggest many future improvements in host range prediction. Whether they will be able to predict which particular strains of bacteria are in a phage’s host range, as can be determined by culture-based methods, is less clear, although the possible applications of receptor-binding protein structure predictions may be a path toward this level of detail. Only time will tell.

## Figures and Tables

**Figure 1 viruses-17-00065-f001:**
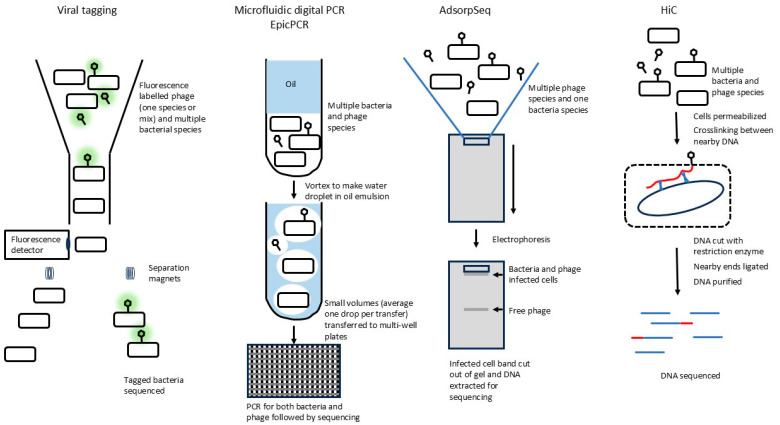
Outline of methods to determine the proximity of phages and hosts. Additional information about each method is found in Table 1 and Section 2.3 in the text. The iPolony method, which is a hybrid method that is principally used to enumerate numbers of phage infected cells, is not shown. Briefly, a mixture of phages and bacteria (some infected) is separated using a FACS machine based on some bacterial property. The selected cells are embedded in a gel matrix, which contains reagents for PCR detection, including degenerate primers for several groups of phages. After PCR, the numbers of cells infected by each type of phage can be determined by differing fluorescent signals or probes.

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
