# Peer review of "Are You My Host? An Overview of Methods Used to Link Bacteriophages with Hosts"

_viruses, 2025, doi:10.3390/v17010065_

Round 1

Reviewer 1 Report

Comments and Suggestions for Authors

The manuscript provides a well-rounded and comprehensive review of the methods used to identify bacteriophage-host relationships, particularly those that dictate host range. I never appreciated the four different “types” of host ranges that exist, and these were clearly explained at the beginning of the manuscript. The author's expertise in the field offers valuable insights into both traditional and modern approaches used for host range determination. The organization of the content into clear categories enhances its readability.

Having been in the phage field for many years, I still learned several new things from reading this review! Quite honestly, I don’t have any significant negative comments or strong suggestions. I did find some terms like "polony" a bit confusing at first, but the explanation was eventually provided in the table. Aesthetically, it might be nice to include some kind of figure or visual since the manuscript is entirely text-based, although I’m not sure what to suggest. Additionally, it could be worth mentioning something about AlphaFold. This AI technology enables scientists to predict/solve the structures of many phage receptor-binding proteins with high confidence, which could enhance host range information derived from genome sequence-based methods. However, this is just a suggestion for the author to consider, not a requirement. Nice job all around.

Reviewer 2 Report

Comments and Suggestions for Authors

This is a welcomed review of various methods that can be used to ascertain host range in phage-bacteria interactions. It is well-written and comprehensive and was interesting to read.  I have a few suggestions:

1)    The conclusions section is somewhat generic and redundant to the introduction.  I would have appreciated more thoughts on how these new methods will or could 

            “evolve” to help answer questions about host range 

2)    For some of the newer methods a figure to visualize how that method works would be nice (and easier to follow than short description in Table 1)

l. 75 A word or two missing in following sentence?: “Penetrative host range is broader than productive host range because there while all phages that kill a bacterium must insert their genome into the host cell, not all bacteria that have a phage genome enter the host cell will produce phage.”

l. 209 “Each one focused on strains of a single species to so(?)it is uncertain if these methods can be applied to less well-defined species of bacteria.”  

Define abbreviations when first used – e.g., FACS

Reviewer 3 Report

Comments and Suggestions for Authors The authors presented here a review of methods that can be used to identify host range of phages, which is crucial to the understanding of phage-host interactions and also phage therapy. The commonly used methods are well characterized as well as the new developments including machine learning specialized predictions of host range. This is suitable for publication after the following concerns are well addressed:
  1. The author classified here the methods discussed into 3 main groups for determining the host range, while most of the discussed including machine learning based predictions and genome sequence based analysis can only be used to predict the host range, the actual determinations and confirmations need to be done through other methods, the authors need to clarify that the methods discussed here are focusing on predictions and determinations, instead of simply saying 'method for determining'
  2. Table 1 is a great conclusion of various methods discussed here, while it's a bit overwhelming and hard to read for readers, it'd be great if the author could find a better way to present the method classes and their specifics and differences, maybe a clear visible figure would be better
  3. Some of the language used needs to be carefully inspected, including line 15 'They also are not', line 89 'they might infect as led to the development'
Comments on the Quality of English Language

needs to be carefully checked
